# SIMULTANEOUS CLASSIFICATION AND OUT-OF-DISTRIBUTION DETECTION USING DEEP NEURAL NETWORKS

## ABSTRACT

Deep neural networks have achieved great success in classification tasks during the last years. However, one major problem to the path towards artificial intelligence is the inability of neural networks to accurately detect samples from novel class distributions and therefore, most of the existent classification algorithms assume that all classes are known prior to the training stage. In this work, we propose a methodology for training a neural network that allows it to efficiently detect out-of-distribution (OOD) examples without compromising much of its classification accuracy on the test examples from known classes. Based on the Outlier Exposure (OE) technique, we propose a novel loss function that achieves state-of-the-art results in out-of-distribution detection with OE both on image and text classification tasks. Additionally, we experimentally show that the combination of our method with the Mahalanobis distance-based classifier achieves state-of-the-art results in the OOD detection task.

## 1 INTRODUCTION

Modern neural networks have recently achieved superior results in classification problems (Krizhevsky et al., 2012; He et al., 2016). However, most of the classification algorithms proposed so far make the assumption that data generated from all the class conditional distributions are available during training time i.e., they make the closed-world assumption. In an open world environment (Bendale & Boult, 2015), where examples from novel class distributions might appear during test time, it is necessary to build classifiers that are able to detect OOD examples while having high classification accuracy on known class distributions.

It is generally known that deep neural networks can make predictions for out-of-distribution (OOD) examples with high confidence (Nguyen et al., 2015). High confidence predictions are undesirable since they consist a symptom of overfitting (Szegedy et al., 2015). They also make the calibration of neural networks difficult. Guo et al. (2017) observed that modern neural networks are miscalibrated by experimentally showing that the average confidence of deep neural networks is usually much higher than their accuracy.

A simple yet effective method to address the problem of the inability of neural networks to detect OOD examples is to train them so that they make highly uncertain predictions for examples generated by novel class distributions. In order to achieve that, Lee et al. (2018a) defined a loss function based on the Kullback-Leibler (KL) divergence metric to minimize the distance between the output distribution given by softmax and the uniform distribution for samples generated by a GAN (Goodfellow et al., 2014). Using a similar loss function, Hendrycks et al. (2019) showed that the technique of Outlier Exposure (OE) that draws anomalies from a real and diverse dataset can outperform the GAN framework for OOD detection.

Using the OE technique, our main contribution is threefold:

- We propose a novel loss function consisting of two regularization terms. The first regularization term minimizes the $l_1$ norm between the output distribution given by softmax and the uniform distribution which constitutes a distance metric between the two distributions (Deza & Deza, 2009). The second regularization term minimizes the Euclidean distance

between the training accuracy of a DNN and its average confidence in its predictions on the training set.

- We experimentally show that the proposed loss function outperforms the previous work of Hendrycks et al. (2019) and achieves state-of-the-art results in OOD detection with OE both on image and text classification tasks.

- We experimentally show that our proposed method can be combined with the Mahalanobis distance-based classifier (Lee et al., 2018b). The combination of the two methods outperforms the original Mahalanobis method in all of the experiments and to the best of our knowledge, achieves state-of-the-art results in the OOD detection task.

## 2    RELATED WORK

Yu et al. (2017) used the GAN framework (Goodfellow et al., 2014) to generate negative instances of seen classes by finding data points that are close to the training instances but are classified as fake by the discriminator. Then, they used those samples in order to train SVM classifiers to detect examples from unseen classes. Similarly, Kliger & Fleishman (2018) used a multi-class GAN framework in order to produce a generator that generates a mixture of nominal data and novel data and a discriminator that performs simultaneous classification and novelty detection.

Hendrycks & Gimpel (2017) proposed a baseline for detecting misclassified and out-of-distibution examples based on their observation that the prediction probability of out-of-distribution examples tends to be lower than the prediction probability for correct examples. Recently, Corbière et al. (2019) also studied the problem of detecting overconfident incorrect predictions. A single-parameter variant of Platt scaling (Platt, 1999), temperature scaling, was proposed by Guo et al. (2017) for calibration of modern neural networks. For image data, based on the idea of Hendrycks & Gimpel (2017), Liang et al. (2018) observed that simultaneous use of temperature scaling and small perturbations at the input can push the softmax scores of in- and out-of-distribution images further apart from each other, making the out-of-distribution images distinguishable. Lee et al. (2018a) generated GAN examples and forced the neural network to have lower confidence in predicting their classes. Hendrycks et al. (2019) substituted the GAN samples with a real and diverse dataset using the technique of OE. Similar works (Malinin & Gales, 2018; Bevandić et al., 2018) also force the model to make uncertain predictions for OOD examples. Using an ensemble of classifiers, Lakshminarayanan et al. (2017) showed that their method was able to express higher uncertainty in OOD examples. Liu et al. (2018) provided theoretical guarantees for detecting OOD examples under the assumption that an upper bound of the fraction of OOD examples is available.

Under the assumption that the pre-trained features of a softmax neural classifier can be fitted well by a class-conditional Gaussian distribution, Lee et al. (2018b) defined a confidence score using the Mahalanobis distance that can efficiently detect abnormal test samples. As also mentioned by Lee et al. (2018b), Euclidean distance can also be used but with less efficiency. We prefer to call these methods Distance-Based Post-Training (DBPT) methods for OOD detection.

## 3    SIMULTANEOUS CLASSIFICATION AND OUT-OF-DISTRIBUTION DETECTION

We consider the multi-class classification problem under the open-world assumption (Bendale & Boult, 2015), where samples from some classes are not available during training. Our task is to design deep neural network classifiers that can achieve high accuracy on examples generated by a learned probability distribution called $D_{in}$ while at the same time, they can effectively detect examples generated by a different probability distribution called $D_{out}$ during the test phase. The examples generated by $D_{in}$ are called in-distribution while the examples generated by $D_{out}$ are called out-of-distribution (OOD). Adopting the idea of Outlier Exposure (OE) proposed by Hendrycks et al. (2019), we train the neural network using training examples sampled from $D_{in}$ and $D_{out}^{OE}$. During the test phase, we evaluate the OOD detection capability of the neural network using examples sampled from $D_{out}^{test}$, where $D_{out}^{OE}$ and $D_{out}^{test}$ are disjoint.

Lee et al. (2018a) and Hendrycks et al. (2019) used the KL divergence metric in order to minimize the distance between the output distribution produced by softmax for the OOD examples and the

uniform distribution. In our work, we choose to minimize the $l_1$ norm between the two distributions which has shown great success in machine learning applications.

Viewing the knowledge of a model as the class conditional distribution it produces over outputs given an input (Hinton et al., 2015), the entropy of this conditional distribution can be used as a regularization method that penalizes confident predictions of a neural network (Pereyra et al., 2017). In our approach, instead of penalizing the confident predictions of posterior probabilities yielded by a neural network, we force it to make predictions for examples generated by $D_{in}$ with an average confidence close to its training accuracy. In such a manner, not only do we make the neural network avoid making overconfident predictions, but we also take into consideration its calibration (Guo et al., 2017).

Let us consider a classification model that can be represented by a parametrized function $f_{\boldsymbol{\theta}}$, where $\boldsymbol{\theta}$ stands for the vector of parameters in $f_{\boldsymbol{\theta}}$. Without loss of generality, assume that the cross entropy loss function is used during training. We propose the following constrained optimization problem for finding $\boldsymbol{\theta}$:

$$
\begin{aligned}
\underset{\boldsymbol{\theta}}{\text{minimize}} \quad & \mathbb{E}_{(x,y)\sim D_{in}}[\mathcal{L}_{CE}(f_{\boldsymbol{\theta}}(x), y)] \\
\text{subject to} \quad & \mathbb{E}_{\mathbf{x}\sim D_{in}}\left[\max_{l=1,\ldots,K}\left(\frac{e^{z_l}}{\sum_{j=1}^{K} e^{z_j}}\right)\right] = A_{tr} \\
& \max_{l=1,\ldots,K}\left(\frac{e^{z_l}}{\sum_{j=1}^{K} e^{z_j}}\right) = \frac{1}{K}, \; \forall x^{(i)} \sim D_{out}^{OE}
\end{aligned}
\tag{1}
$$

where $\mathcal{L}_{CE}$ is the cross entropy loss function and $K$ is the number of classes available in $D_{in}$. Even though the constrained optimization problem (1) can be used for training various classification models, for clarity we limit our discussion to deep neural networks. Let $\mathbf{z}$ denote the vector representation of the example $x^{(i)}$ in the feature space produced by the last layer of the deep neural network (DNN) and let $A_{tr}$ be the training accuracy of the DNN. Observe that the optimization problem (1) minimizes the cross entropy loss function subject to two additional constraints. The first constraint forces the average maximum prediction probabilities calculated by the softmax layer towards the training accuracy of the DNN for examples sampled from $D_{in}$, while the second constraint forces the maximum probability calculated by the softmax layer towards $\frac{1}{K}$ for all examples sampled from the probability distribution $D_{out}^{OE}$. In other words, the first constraint makes the DNN predict examples from known classes with an average confidence close to its training accuracy, while the second constraint forces the DNN to be highly uncertain for examples of classes it has never seen before by producing a uniform distribution at the output for examples sampled from the probability distribution $D_{out}^{OE}$. It is also worth noting that the first constraint of (1) uses the training accuracy of the neural network $A_{tr}$ which is not available in general. To handle this issue, one can train a neural network by only minimizing the cross entropy loss function for a few number of epochs in order to calculate $A_{tr}$ and then fine-tune it using (1).

Because solving the nonconvex constrained optimization problem described by (1) is extremely difficult, let us introduce Lagrange multipliers (Boyd & Vandenberghe, 2004) and convert it into the following unconstrained optimization problem:

$$
\begin{aligned}
\underset{\boldsymbol{\theta}}{\text{minimize}} \quad & \mathbb{E}_{(x,y)\sim D_{in}}[\mathcal{L}_{CE}(f_{\boldsymbol{\theta}}(x), y)] + \lambda_1\left(A_{tr} - \mathbb{E}_{\mathbf{x}\sim D_{in}}\left[\max_{l=1,\ldots,K}\left(\frac{e^{z_l}}{\sum_{j=1}^{K} e^{z_j}}\right)\right]\right) \\
& + \lambda_2 \sum_{x^{(i)}\sim D_{out}^{OE}}\left(\frac{1}{K} - \max_{l=1,\ldots,K}\left(\frac{e^{z_l}}{\sum_{j=1}^{K} e^{z_j}}\right)\right)
\end{aligned}
\tag{2}
$$

where it is worth mentioning that in (2), we used only one Lagrange multiplier for the second set of constraints in (1) instead of using one for each constraint in order to avoid introducing a large number of hyperparameters to our loss function. This modification is a special case where we consider the Lagrange multiplier $\lambda_2$ to be common for each individual constraint involving a different $x^{(i)} \sim D_{out}^{OE}$. Note also that according to the original Lagrangian theory, one should

optimize the objective function of (2) both with respect to $\boldsymbol{\theta}, \lambda_1$ and $\lambda_2$ but as it commonly happens in machine learning applications, we approximate the original problem by calculating appropriate values for $\lambda_1$ and $\lambda_2$ through a validation technique (Hastie et al., 2001).

After converting the constrained optimization problem (1) into an unconstrained optimization problem as described by (2), it is possible that at each training epoch, the maximum prediction probability produced by softmax for each example drawn from $D_{out}^{OE}$ changes, introducing difficulties in making the DNN produce a uniform distribution at the output for those examples. For instance, assume that we have a $K$-class classifier with $K = 3$ and at epoch $t_n$, the maximum prediction probability produced by softmax for an example $x^{(i)} \sim D_{out}^{OE}$ corresponds to the second class. Then, the last term of (2) will push the prediction probability of example $x^{(i)}$ for the second class towards $\frac{1}{3}$ while concurrently increasing the prediction probabilities for either the first class or the third class or both. At the next epoch $t_{n+1}$, it is possible that the prediction probability for either the first class or the third class becomes the maximum among the three and hence, the last term of (2) will push that one towards $\frac{1}{3}$ by possibly increasing again the prediction probability for the second class. It becomes obvious that this process introduces difficulties in making the DNN produce a uniform distribution at the output for examples sampled from $D_{out}^{OE}$. However, this issue can be resolved by concurrently pushing all the prediction probabilities produced by the softmax layer for examples drawn from $D_{out}^{OE}$ towards $\frac{1}{K}$.

Additionally, in order to prevent the second and the third term of (2) from taking negative values during training, let us convert (2) into the following:

$$
\begin{aligned}
\underset{\boldsymbol{\theta}}{\text{minimize}} \ \ &\mathbb{E}_{(x,y)\sim D_{in}}[\mathcal{L}_{CE}(f_{\boldsymbol{\theta}}(x), y)] + \lambda_1 \left( A_{tr} - \mathbb{E}_{x \sim D_{in}} \left[ \max_{l=1,\ldots,K} \left( \frac{e^{z_l}}{\sum_{j=1}^{K} e^{z_j}} \right) \right] \right)^2 \\
&+ \lambda_2 \sum_{x^{(i)} \sim D_{out}^{OE}} \sum_{l=1}^{K} \left| \frac{1}{K} - \frac{e^{z_l}}{\sum_{j=1}^{K} e^{z_j}} \right|
\end{aligned}
\tag{3}
$$

The second term of the the loss function described by (3) minimizes the squared distance between the training accuracy of the DNN and the average confidence in its predictions for examples drawn from $D_{in}$. Additionally, the third term of (3) minimizes the $l_1$ norm between the uniform distribution and the distribution produced by the softmax layer for the examples drawn from $D_{out}^{OE}$.

While converting the unconstrained optimization problem (2) into (3), one could use several combinations of norms to minimize. However, we found that minimizing the squared distance between the training accuracy of the DNN and the average confidence in its predictions for examples drawn from $D_{in}$ and the $l_1$ norm between the uniform distribution and the distribution produced by the softmax layer for the examples drawn from $D_{out}^{OE}$ works best. This is because $l_1$ norm uniformly attracts all the prediction probabilities produced by softmax to the desired value $\frac{1}{K}$, better contributing to producing a uniform distribution at the output of the DNN for the examples drawn from $D_{out}^{OE}$. On the other hand, minimizing the squared distance between the training accuracy of the DNN and the average confidence in its predictions for examples drawn from $D_{in}$ emphasizes more on attracting the maximum softmax probabilities that are further away from the average confidence of the DNN, making the neural network better detect in- and out-of-distribution examples at the low softmax probability levels.

## 4 EXPERIMENTS

During the experiments, we observed that if we start training the DNN with a relatively high value of $\lambda_1$, the learning process might slow down since we constantly force the neural network to make predictions with an average confidence close to its training accuracy. Therefore, it is recommended to split the training of the algorithm into two stages where in the first stage, we train the DNN using only the cross entropy loss function until it reaches the desired level of accuracy $A_{tr}$ and then using a fixed $A_{tr}$, we fine-tune it using the combined loss function given by (3).

### 4.1 COMPARISON WITH STATE-OF-THE-ART IN OE

The experimental setting is as follows. We draw samples from $D_{in}$ and we train the DNN until it reaches the desired level of accuracy $A_{tr}$. Then, drawing samples from $D_{out}^{OE}$, we fine-tune it using the combined loss function given by (3). During the test phase, we evaluate the OOD detection capability of the DNN using examples from $D_{out}^{test}$ which is disjoint from $D_{out}^{OE}$. We demonstrate the effectiveness of our method in both image and text classification tasks by comparing it with the previous OOD detection with OE method proposed by Hendrycks et al. (2019). A part of our experiments was based on the publicly available code of Hendrycks et al. (2019).

#### 4.1.1 EVALUATION METRICS

Our OOD detection method belongs to the class of Maximum Softmax Probability (MSP) detectors (Hendrycks & Gimpel, 2017) and therefore, we adopt the evaluation metrics used in Hendrycks et al. (2019). Defining the OOD examples as the positive class and the in-distribution examples as the negative class, the performance metrics associated with OOD detection are the following:

- False Positive Rate at $N\%$ True Positive Rate (*FPRN*): This performance metric (Balntas et al., 2016; Kumar et al., 2016) measures the capability of an OOD detector when the maximum softmax probability threshold is set to a predefined value. More specifically, assuming $N\%$ of OOD examples need to be detected during the test phase, we calculate a threshold in the softmax probability space and given that threshold, we measure the false positive rate, i.e. the ratio of in-distribution examples that are incorrectly classified as OOD.

- Area Under the Receiver Operating Characteristic curve (AUROC): In the out-of-distribution detection task, the ROC curve (Davis & Goadrich, 2006) summarizes the performance of an OOD detection method for varying threshold values.

- Area Under the Precision-Recall curve (AUPR): The AUPR (Manning & Schütze, 1999) is an important measure when there exists a class-imbalance between OOD and in-distribution examples in a dataset. As in Hendrycks et al. (2019), in our experiments, the ratio of OOD and in-distribution test examples is 1:5.

#### 4.1.2 IMAGE CLASSIFICATION EXPERIMENTS

**Results.** The results of the image classification experiments are shown in Table 1. In Figure 1, as an example, we plot the histogram of softmax probabilities using CIFAR-10 as $D_{in}$ and Places365 as $D_{out}^{test}$. The detailed description of the image datasets used in the image OOD detection experiments is presented in Appendix A.2.

**Network Architecture and Training Details.** Similar to Hendrycks et al. (2019), for CIFAR 10 and CIFAR 100 experiments, we used 40-2 wide residual networks (WRNs) proposed by Zagoruyko & Komodakis (2016). We initially trained the WRN for 100 epochs using a cosine learning rate (Loshchilov & Hutter, 2017) with an initial value 0.1, a dropout rate of 0.3 and a batch size of 128. As in Hendrycks et al. (2019), we also used Nesterov momentum and $l_2$ weight regularization with a decay factor of 0.0005. For CIFAR 10, we fine-tuned the network for 15 epochs minimizing the loss function given by (3) using a learning rate of 0.001, while for the CIFAR 100 the corresponding number of epochs was 20. For the SVHN experiments, we trained 16-4 WRNs using a learning rate of 0.01, a dropout rate of 0.4 and a batch size of 128. We then fine-tuned the network for 5 epochs using a learning rate of 0.001. During fine-tuning, the 80 Million Tiny Images dataset was used as $D_{out}^{OE}$. The values of the hyperparameters $\lambda_1$ and $\lambda_2$ were chosen in the range $[0.03, 0.09]$ using a separate validation dataset $D_{out}^{val}$ similar to Hendrycks et al. (2019). We note that $D_{out}^{val}$ and $D_{out}^{test}$ are disjoint. The data used for validation are presented in Appendix A.3.

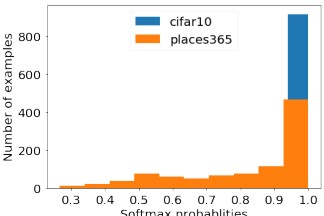

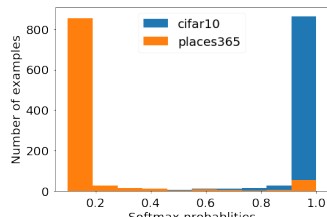

Figure 1: Histogram of softmax probabilities with CIFAR-10 as $D_{in}$ and Places365 as $D_{out}^{test}$ (1,000 samples from each dataset). *Top*: MSP baseline detector. *Bottom*: MSP detector fine-tuned with (3).

|  | FPR95↓ | | AUROC↑ | | AUPR↑ | |
|---|---|---|---|---|---|---|
| $D_{in}$ | +OE | OURS | +OE | OURS | +OE | OURS |
| SVHN | 0.10 | **0.03** | 99.98 | **99.99** | **99.83** | 99.55 |
| CIFAR-10 | 9.50 | **6.56** | 97.81 | **98.40** | 90.48 | **93.08** |
| CIFAR-100 | 38.50 | **28.89** | 87.89 | **91.80** | 58.15 | **71.50** |

Table 1: Image OOD example detection for the maximum softmax probability (MSP) baseline detector after fine-tuning with OE (Hendrycks et al., 2019) versus fine-tuning with our proposed loss function given by (3). All results are percentages and averaged over 10 runs and over 8 OOD datasets. Detailed experimental results are in Appendix A.1.

**Contribution of each regularization term.** To demonstrate the effect of each regularization term of the loss function described by (3) in the OOD detection task, we ran some additional image classification experiments which are presented in Table 2. For these experiments, we incrementally added each regularization term to the loss function described by (3) and we measured its effect both in the OOD detection evaluation metrics as well as in the accuracy of the DNN on the test images of $D_{in}$. The results of these experiments validate that the combination of the two regularization terms of (3) not only improves the OOD detection performance of the DNN but also improves its accuracy on the test examples of $D_{in}$ compared to the case where $\lambda_1 = 0$. Table 2 also demonstrates that our method can significantly improve the OOD detection performance of the DNN compared to the case where only the cross-entropy loss is minimized at the expense of only an insignificant degradation in the test accuracy of the DNN on examples generated by $D_{in}$.

| $D_{in}$ | $\lambda_1$ | $\lambda_2$ | FPR95↓ | AUROC↑ | AUPR↑ | Test Accuracy($D_{in}$) |
|---|---|---|---|---|---|---|
| | - | - | 34.94 | 89.27 | 59.16 | 94.65 |
| CIFAR-10 | - | ✓ | 8.87 | 96.72 | 77.65 | 92.72 |
| | ✓ | ✓ | 6.56 | 98.40 | 93.08 | 93.86 |
| | - | - | 62.66 | 73.11 | 30.05 | 75.73 |
| CIFAR-100 | - | ✓ | 26.75 | 91.59 | 68.27 | 71.29 |
| | ✓ | ✓ | 28.89 | 91.80 | 71.50 | 73.14 |

Table 2: Contribution of each regularization term of (3) on the OOD detection performance and the test accuracy of the DNN. Results are averaged over 10 runs and over 8 OOD datasets.

### 4.1.3 TEXT CLASSIFICATION EXPERIMENTS

**Results.** The results of the text classification experiments are shown in Table 3. The detailed description of the text datasets used in the NLP OOD detection experiments is presented in Appendix B.1.

|  | FPR90↓ | | AUROC↑ | | AUPR↑ | |
|---|---|---|---|---|---|---|
| $D_{in}$ | +OE | OURS | +OE | OURS | +OE | OURS |
| 20 Newsgroups | 4.86 | **0.63** | 97.71 | **99.18** | 91.91 | **97.02** |
| TREC | 0.78 | **0.75** | 99.28 | **99.32** | **97.64** | 97.52 |
| SST | 27.33 | **17.91** | 89.27 | **93.79** | 59.23 | **74.10** |

Table 3: NLP OOD example detection for the maximum softmax probability (MSP) baseline detector after fine-tuning with OE (Hendrycks et al., 2019) versus fine-tuning with our proposed loss function given by (3). All results are percentages and averaged over 10 runs and over 10 OOD datasets. Detailed experimental results are in Appendix B.3.

**Network Architecture and Training Details.** For all text classification experiments, similar to Hendrycks et al. (2019), we train 2-layer GRUs (Cho et al., 2014) for 5 epochs with learning rate 0.01 and a batch size of 64 and then we fine-tune them for 2 epochs using the loss function given by (3). During fine-tuning, the WikiText-2 dataset was used as $D_{out}^{OE}$. The values of the hyperparameters

$\lambda_1$ and $\lambda_2$ were chosen in the range $[0.04, 0.1]$ using a separate validation dataset as described in Appendix B.2.

## 4.2 A Combination of OE and DBPT Methods for OOD Detection

Lee et al. (2018b) proposed a DBPT method for OOD detection that can be applied to any pre-trained softmax neural classifier. Under the assumption that the pre-trained features of a DNN can be fitted well by a class-conditional Gaussian distribution, they defined the confidence score using the Mahalanobis distance with respect to the closest class-conditional probability distribution, where its parameters are chosen as empirical class means and tied empirical covariance of training samples (Lee et al., 2018b). To further distinguish in- and out-of-distribution examples, they proposed two additional techniques. In the first technique, they added a small perturbation before processing each input example to increase the confidence score of their method. In the second technique, they proposed a feature ensemble method in order to obtain a better calibrated score. The feature ensemble method extracts all the hidden features of the DNN and computes their empirical class mean and tied covariances. Subsequently, it calculates the Mahalanobis distance-based confidence score for each layer and finally calculates the weighted average of these scores by training a logistic regression detector using validation samples in order to calculate the weight of each layer at the final confidence score.

Since the Mahalanobis distance-based classifier proposed by Lee et al. (2018b) is a post-training method, it can be combined with our proposed loss function described by (3). More specifically, in our experiments, we initially trained a DNN using the standard cross entropy loss function and then we fine-tuned it with the proposed loss function given by (3). After fine-tuning, we applied the Mahalanobis distance-based classifier and we compared the obtained results against the results presented in Lee et al. (2018b). The simulation experiments on image classification tasks show that the combination of our method which belongs to the OE "family" of methods and the Mahalanobis distance-based classifier which belongs to the "family" of DBPT methods achieves state-of-the-art results in the OOD detection task. A part of our experiments was based on the publicly available code of Lee et al. (2018b).

### 4.2.1 Evaluation Metrics

To demonstrate the adaptability of our method, in these experiments, we adopt the OOD detection evaluation metrics used in Lee et al. (2018b).

- True Negative Rate at $N\%$ True Positive Rate (*TNRN*): This performance metric measures the capability of an OOD detector to detect true negative examples when the true positive rate is set to $95\%$.

- Area Under the Receiver Operating Characteristic curve (AUROC): In the out-of-distribution detection task, the ROC curve (Davis & Goadrich, 2006) summarizes the performance of an OOD detection method for varying threshold values.

- Detection Accuracy (DAcc): As also mentioned in Lee et al. (2018b), this evaluation metric corresponds to the maximum classification probability over all possible thresholds $\epsilon$:

$$1 - \min_{\epsilon}\{D_{in}(q(\boldsymbol{x}) \leq \epsilon)P(\boldsymbol{x} \text{ is from } D_{in}) + D_{out}(q(\boldsymbol{x}) > \epsilon)P(\boldsymbol{x} \text{ is from } D_{out})\},$$

  where $q(\boldsymbol{x})$ is a confidence score. Similar to Lee et al. (2018b), we assume that $P(\boldsymbol{x} \text{ is from } D_{in}) = P(\boldsymbol{x} \text{ is from } D_{out})$.

### 4.2.2 Experimental Setup

To demonstrate the adaptability and the effectiveness of our method, we adopt the experimental setup of Lee et al. (2018b). We train ResNet (He et al., 2016) with 34 layers using CIFAR-10, CIFAR-100 and SVHN datasets as $D_{in}$. For the CIFAR experiments, SVHN, TinyImageNet (a sample of 10,000 images drawn from the ImageNet dataset) and LSUN are used as $D_{out}^{test}$. For the SVHN experiments, CIFAR-10, TinyImageNet and LSUN are used as $D_{out}^{test}$. Both TinyImageNet and LSUN images are downsampled to $32 \times 32$.

Similar to Lee et al. (2018b), for the Mahalanobis distance-based classifier, we train the ResNet model for 200 epochs with batch size 128 by minimizing the cross entropy loss using the SGD

| $D_{in}$ | $D_{out}^{test}$ | TNR95↑ | | AUROC↑ | | DAcc↑ | |
|---|---|---|---|---|---|---|---|
| | | Mahal. | OURS+Mahal. | Mahal. | OURS+Mahal. | Mahal. | OURS+Mahal. |
| | SVHN | 96.4 | **97.3** | 99.1 | **99.2** | 95.8 | **96.3** |
| CIFAR-10 | TinyImageNet | 97.1 | **98.8** | 99.5 | **99.6** | 96.3 | **97.3** |
| | LSUN | 98.9 | **99.7** | 99.7 | **99.8** | 97.7 | **98.5** |
| | SVHN | 91.9 | **93.0** | 98.4 | **98.7** | 93.7 | **94.2** |
| CIFAR-100 | TinyImageNet | 90.9 | **92.3** | 98.2 | **98.3** | 93.3 | **93.9** |
| | LSUN | 90.9 | **95.6** | 98.2 | **98.6** | 93.5 | **95.4** |
| | CIFAR-10 | 98.4 | **99.9** | 99.3 | **99.9** | 96.9 | **99.2** |
| SVHN | TinyImageNet | 99.9 | **100.0** | 99.9 | **100.0** | 99.1 | **99.9** |
| | LSUN | 99.9 | **100.0** | 99.9 | **100.0** | 99.5 | **100.0** |

Table 4: Comparison between the Mahalanobis distance-based classifier (Lee et al., 2018b) and the combination of our proposed method with the Mahalanobis method. The hyper-parameters are tuned using a validation dataset of in- and out-of-distribution data similar to Lee et al. (2018b). Additional training details for our method are presented in Appendix C.

algorithm with momentum 0.9. The learning rate starts at 0.1 and is dropped by a factor of 10 at 50% and 75% of the training progress, respectively. Subsequently, we compute the Mahalanobis distance-based confidence score using both the input pre-processing and the feature ensemble techniques. The hyper-parameters that need to be tuned are the magnitude of the noise added at each test input example as well as the layer indexes for feature ensemble. Similar to Lee et al. (2018b), both of them are tuned using a separate validation dataset consisting of both in- and out-of-distribution data.

Since the Mahalanobis distance-based classifier belongs to the "family" of DBPT methods for OOD detection tasks, it can be combined with our proposed method. More specifically, we initially train the ResNet model with 34 layers for 200 epochs using exactly the same training details as mentioned above. Subsequently, we fine-tune the network with the proposed loss function described by (3) using the 80 Million Tiny Images as $D_{out}^{OE}$. During fine-tuning, we use the SGD algorithm with momentum 0.9 and a cosine learning rate (Loshchilov & Hutter, 2017) with an initial value 0.001 using a batch size of 128 for data sampled from $D_{in}$ and a batch size of 256 for data sampled from $D_{out}^{OE}$. For CIFAR-10 and 100 experiments, we fine-tuned the network for 30 and 20 epochs respectively, while for SVHN the corresponding number of epochs was 5. The values of the hyper-parameters $\lambda_1$ and $\lambda_2$ were chosen using a separate validation dataset consisting of both in- and out-of-distribution images similar to Lee et al. (2018b). The results are shown in Table 4.

**Discussion.** The results in Table 4 demonstrate the effectiveness of our method when combined with the Mahalanobis distance-based classifier since it outperforms the original version of the Mahalanobis method proposed by Lee et al. (2018b) in all of the experiments. This result validates the contribution of our technique further, since it does not only achieve state-of-the-art results in OOD detection with OE, but it can be additionally combined with DBPT methods like the Mahalanobis distance-based classifier to achieve state-of-the-art results in the OOD detection task. The superior performance of our method when combined with the Mahalanobis distance-based classifier can be justified by the fact that the latter extracts the learned features from the layer(s) of the DNN and it subsequently uses those features to define a confidence score based on the Mahalanobis distance. The simulation results presented in Table 1 and Table 3 showed that our method can teach the DNN to learn feature representations that can further distinguish in- and out-of distribution data and therefore, the combination of the two methods improves the OOD detection capability of a DNN.

## 5 CONCLUSION

In this paper, we proposed a method for simultaneous classification and out-of-distribution detection. The proposed loss function includes two regularization terms where the first minimizes the $l_1$ norm between the output distribution of the softmax layer of a DNN and the uniform distribution, while the second minimizes the Euclidean distance between the training accuracy of a DNN and its average confidence in its predictions on the training set. Experimental results showed that the proposed loss function achieves state-of-the-art results in OOD detection with OE (Hendrycks et al., 2019) in both image and text classification tasks. Additionally, we experimentally showed that our method can be combined with DBPT methods for OOD detection like the Mahalanobis distance-based classifier (Lee et al., 2018b) and achieves state-of-the-art results in the OOD detection task.

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

# A EXPANDED IMAGE OOD DETECTION RESULTS AND DATASETS USED FOR COMPARISON WITH STATE-OF-THE-ART IN OE

## A.1 IMAGE OOD DETECTION RESULTS

| $D_{in}$ | $D_{out}^{test}$ | FPR95↓ | | AUROC↑ | | AUPR↑ | |
|---|---|---|---|---|---|---|---|
| | | +OE | OURS | +OE | OURS | +OE | OURS |
| SVHN | Gaussian | 0.0 | 0.0 | 100. | 100. | 100. | 99.4 |
| | Bernulli | 0.0 | 0.0 | 100. | 100. | 100. | 99.2 |
| | Blobs | 0.0 | 0.0 | 100. | 100. | 100. | 99.6 |
| | Icons-50 | 0.3 | 0.1 | 99.8 | 99.9 | 99.2 | 99.5 |
| | Textures | 0.2 | 0.1 | 100. | 100. | 99.7 | 99.6 |
| | Places365 | 0.1 | 0.0 | 100. | 100. | 99.9 | 99.7 |
| | LSUN | 0.1 | 0.0 | 100. | 100. | 99.9 | 99.7 |
| | CIFAR-10 | 0.1 | 0.0 | 100. | 100. | 99.9 | 99.7 |
| | Mean | 0.10 | **0.03** | 99.98 | **99.99** | **99.83** | 99.55 |
| CIFAR-10 | Gaussian | 0.7 | 0.7 | 99.6 | 99.8 | 94.3 | 99.0 |
| | Rademacher | 0.5 | 1.1 | 99.8 | 99.6 | 97.4 | 97.6 |
| | Blobs | 0.6 | 1.5 | 99.8 | 99.1 | 98.9 | 91.7 |
| | Textures | 12.2 | 4.0 | 97.7 | 98.9 | 91.0 | 95.0 |
| | SVHN | 4.8 | 1.4 | 98.4 | 99.6 | 89.4 | 97.9 |
| | Places365 | 17.3 | 13.3 | 96.2 | 96.9 | 87.3 | 89.5 |
| | LSUN | 12.1 | 6.7 | 97.6 | 98.4 | 89.4 | 91.9 |
| | CIFAR-100 | 28.0 | 23.8 | 93.3 | 94.9 | 76.2 | 82.0 |
| | Mean | 9.50 | **6.56** | 97.81 | **98.40** | 90.48 | **93.08** |
| CIFAR-100 | Gaussian | 12.1 | 0.7 | 95.7 | 99.7 | 71.1 | 97.2 |
| | Rademacher | 17.1 | 0.7 | 93.0 | 99.7 | 56.9 | 96.2 |
| | Blobs | 12.1 | 1.3 | 97.2 | 99.6 | 86.2 | 96.3 |
| | Textures | 54.4 | 50.1 | 84.8 | 87.8 | 56.3 | 61.5 |
| | SVHN | 42.9 | 16.7 | 86.9 | 94.9 | 52.9 | 74.1 |
| | Places365 | 49.8 | 47.8 | 86.5 | 88.1 | 57.9 | 58.5 |
| | LSUN | 57.5 | 56.6 | 83.4 | 85.9 | 51.4 | 53.0 |
| | CIFAR-10 | 62.1 | 57.2 | 75.7 | 78.7 | 32.6 | 35.2 |
| | Mean | 38.50 | **28.89** | 87.89 | **91.80** | 58.15 | **71.50** |

Table 5: Image OOD example detection for the maximum softmax probability (MSP) baseline detector after fine-tuning with OE (Hendrycks et al., 2019) versus fine-tuning with our proposed loss function given by (3). All results are percentages and averaged over 10 runs. Values are rounded to the first decimal digit.

## A.2 $D_{in}$, $D_{out}^{OE}$ AND $D_{out}^{test}$ FOR IMAGE EXPERIMENTS

**SVHN:** The Street View House Number (SVHN) dataset (Netzer et al., 2011) consists of $32 \times 32$ color images out of which 604,388 are used for training and 26,032 are used for testing. The dataset has 10 classes and was collected from real Google Street View images. Similar to Hendrycks et al. (2019), we rescale the pixels of the images to be in $[0, 1]$.

**CIFAR 10:** This dataset (Krizhevsky & Hinton, 2009) contains 10 classes and consists of 60,000 $32 \times 32$ color images out of which 50,000 belong to the training and 10,000 belong to the test set. Before training, we standardize the images per channel similar to Hendrycks et al. (2019).

**CIFAR 100:** This dataset (Krizhevsky & Hinton, 2009) consists of 20 distinct superclasses each of which contains 5 different classes giving us a total of 100 classes. The total number of images in the dataset are 60,000 and we use the standard 50,000/10,000 train/test split. Before training, we standardize the images per channel similar to Hendrycks et al. (2019).

**80 Million Tiny Images:** The 80 Million Tiny Images dataset (Torralba et al., 2008) was exclusively used in our experiments in order to represent $D_{out}^{OE}$. It consists of $32 \times 32$ color images collected from the Internet. Similar to Hendrycks et al. (2019), in order to make sure that $D_{out}^{OE}$ and $D_{out}^{test}$ are

disjoint, we removed all the images of the dataset that appear on CIFAR 10 and CIFAR 100 datasets.

**Places365:** Places365 dataset introduced by Zhou et al. (2018) was exclusively used in our experiments in order to represent $D_{out}^{test}$. It consists of millions of photographs of scenes.

**Gaussian:** A synthetic image dataset created by i.i.d. sampling from an isotropic Gaussian distribution.

**Bernoulli:** A synthetic image dataset created by sampling from a Bernoulli distribution.

**Blobs:** A synthetic dataset of images with definite edges.

**Icons-50:** This dataset intoduced by Hendrycks & Dietterich (2018) consists of 10,000 images belonging to 50 classes of icons. As part of preprocessing, we removed the class "Number" in order to make it disjoint from the SVHN dataset.

**Textures:** This dataset contains 5,640 textural images (Cimpoi et al., 2014).

**LSUN:** It consists of around 1 million large-scale images of scenes (Yu et al., 2015).

**Rademacher:** A synthetic image dataset created by sampling from a symmetric Rademacher distribution.

### A.3 VALIDATION DATA FOR IMAGE EXPERIMENTS

**Uniform Noise:** A synthetic image dataset where each pixel is sampled from $\mathcal{U}[0, 1]$ or $\mathcal{U}[-1, 1]$ depending on the input space of the classifier.

**Arithmetic Mean:** A synthetic image dataset created by randomly sampling a pair of in-distribution images and subsequently taking their pixelwise arithmetic mean.

**Geometric Mean:** A synthetic image dataset created by randomly sampling a pair of in-distribution images and subsequently taking their pixelwise geometric mean.

**Jigsaw:** A synthetic image dataset created by partitioning an image sampled from $D_{in}$ into 16 equally sized patches and by subsequently permuting those patches.

**Speckle Noised:** A synthetic image dataset created by applying speckle noise to images sampled from $D_{in}$.

**Inverted Images:** A synthetic image dataset created by shifting and reordering the color channels of images sampled from $D_{in}$.

**RGB Ghosted:** A synthetic image dataset created by inverting the color channels of images sampled from $D_{in}$.

## B EXPANDED TEXT OOD DETECTION RESULTS AND DATASETS USED FOR COMPARISON WITH STATE-OF-THE-ART IN OE

### B.1 $D_{in}$, $D_{out}^{OE}$ AND $D_{out}^{test}$ FOR NLP EXPERIMENTS

**20 Newsgroups:** This dataset contains 20 different newsgroups, each corresponding to a specific topic. It contains around 19,000 examples and we used the standard 60/40 train/test split.

**TREC:** A question classification dataset containing around 6,000 examples from 50 different classes. Similar to Hendrycks et al. (2019), we used 500 examples for the test phase and the rest for training.

**SST:** The Stanford Sentiment Treebank (Socher et al., 2013) is a binary classification dataset for sentiment prediction of movie reviews containing around 10,000 examples.

**WikiText-2:** This dataset contains over 2 million articles from Wikipedia and is exclusively used as $D_{out}^{OE}$ in our experiments. We used the same preprocessing as in Hendrycks et al. (2019) in order to have a valid comparison.

**SNLI:** The Stanford Natural Language Inference (SNLI) corpus is a collection of 570,000 human-written English sentence pairs (Bowman et al., 2015).

**IMDB:** A sentiment classification dataset containing movies reviews.

**Multi30K:** A dataset of English and German descriptions of images (Elliott et al., 2016). For our experiments, only the English descriptions were used.

**WMT16:** A dataset used for machine translation tasks. For our experiments, only the English part of the test set was used.

**Yelp:** A dataset containing reviews of users for businesses on Yelp.

**EWT:** The English Web Treebank (EWT) consists of 5 different datasets: weblogs (EWT-W), news-groups (EWT-N), emails (EWT-E), reviews (EWT-R) and questions-answers (EWT-A).

## B.2 VALIDATION DATA FOR NLP EXPERIMENTS

The validation dataset $D_{out}^{val}$ used for the NLP OOD detection experiments was constructed as follows. For each $D_{in}$ dataset used, we used the rest two in-distribution datasets as $D_{out}^{val}$. For instance, during the experiments where *20 Newsgroups* represented $D_{in}$, we used *TREC* and *SST* as $D_{out}^{val}$ making sure that $D_{out}^{val}$ and $D_{out}^{test}$ are disjoint.

## B.3 TEXT OOD DETECTION RESULTS

| $D_{in}$ | $D_{out}^{test}$ | FPR90↓ +OE | FPR90↓ OURS | AUROC↑ +OE | AUROC↑ OURS | AUPR↑ +OE | AUPR↑ OURS |
|---|---|---|---|---|---|---|---|
| 20 Newsgroups | SNLI | 12.5 | 2.1 | 95.1 | 97.1 | 86.3 | 93.0 |
| | IMDB | 18.6 | 2.5 | 93.5 | 98.2 | 74.5 | 92.9 |
| | Multi30K | 3.2 | 0.1 | 97.3 | 99.4 | 93.7 | 98.6 |
| | WMT16 | 2.0 | 0.2 | 98.8 | 99.8 | 96.1 | 99.4 |
| | Yelp | 3.9 | 0.4 | 97.8 | 99.6 | 87.9 | 97.9 |
| | EWT-A | 1.2 | 0.2 | 99.2 | 99.8 | 97.3 | 98.4 |
| | EWT-E | 1.4 | 0.1 | 99.2 | 99.9 | 97.2 | 98.9 |
| | EWT-N | 1.8 | 0.5 | 98.7 | 99.2 | 95.7 | 94.5 |
| | EWT-R | 1.7 | 0.1 | 98.9 | 99.4 | 96.6 | 98.3 |
| | EWT-W | 2.4 | 0.1 | 98.5 | 99.4 | 93.8 | 98.3 |
| | Mean | 4.86 | **0.63** | 97.71 | **99.18** | 91.91 | **97.02** |
| TREC | SNLI | 4.2 | 0.8 | 98.1 | 99.1 | 91.6 | 94.9 |
| | IMDB | 0.6 | 0.6 | 99.4 | 98.9 | 97.8 | 97.1 |
| | Multi30K | 0.3 | 0.2 | 99.7 | 99.9 | 99.0 | 99.6 |
| | WMT16 | 0.2 | 0.2 | 99.8 | 99.9 | 99.4 | 99.6 |
| | Yelp | 0.4 | 0.8 | 99.7 | 99.1 | 96.1 | 92.9 |
| | EWT-A | 0.9 | 4.0 | 97.7 | 98.0 | 96.1 | 95.6 |
| | EWT-E | 0.4 | 0.3 | 99.5 | 99.2 | 99.1 | 98.1 |
| | EWT-N | 0.3 | 0.2 | 99.6 | 99.9 | 99.2 | 99.6 |
| | EWT-R | 0.4 | 0.2 | 99.5 | 99.6 | 98.8 | 98.9 |
| | EWT-W | 0.2 | 0.2 | 99.7 | 99.6 | 99.4 | 98.9 |
| | Mean | 0.78 | **0.75** | 99.28 | **99.32** | **97.64** | 97.52 |
| SST | SNLI | 33.4 | 7.4 | 86.8 | 95.8 | 52.0 | 76.4 |
| | IMDB | 32.6 | 10.8 | 85.9 | 95.8 | 51.5 | 77.6 |
| | Multi30K | 33.0 | 5.1 | 88.3 | 97.9 | 58.9 | 86.9 |
| | WMT16 | 17.1 | 3.6 | 92.9 | 98.3 | 68.8 | 88.1 |
| | Yelp | 11.3 | 15.6 | 92.7 | 95.2 | 60.0 | 81.1 |
| | EWT-A | 33.6 | 21.4 | 87.2 | 92.7 | 53.8 | 70.8 |
| | EWT-E | 26.5 | 22.6 | 90.4 | 92.4 | 63.7 | 67.7 |
| | EWT-N | 27.2 | 19.2 | 90.1 | 93.6 | 62.0 | 67.4 |
| | EWT-R | 41.4 | 36.7 | 85.6 | 88.1 | 54.7 | 62.5 |
| | EWT-W | 17.2 | 36.7 | 92.8 | 88.1 | 66.9 | 62.5 |
| | Mean | 27.33 | **17.91** | 89.27 | **93.79** | 59.23 | **74.10** |

Table 6: NLP OOD example detection for the maximum softmax probability (MSP) baseline detector after fine-tuning with OE (Hendrycks et al., 2019) versus fine-tuning with our proposed loss function given by (3). All results are percentages and the result of 10 runs. Values are rounded to the first decimal digit.

## C  TRAINING DETAILS FOR THE EXPERIMENTAL RESULTS FOR COMPARISON WITH MAHALANOBIS DISTANCE-BASED CLASSIFIER

During fine-tuning with our proposed loss function given by (3), we used the training details presented in Table 7. The values of the hyper-parameters $\lambda_1$ and $\lambda_2$ were chosen using a separate validation dataset consisting of both in- and out-of-distribution images similar to Lee et al. (2018b).

| $D_{in}$ | $\lambda_1$ | $\lambda_2$ | # Epochs | Test Accuracy($D_{in}$) |
|---|---|---|---|---|
| CIFAR-10 | 0.15 | 0.15 | 30 | 94.60 |
| CIFAR-100 | 0.09 | 0.07 | 20 | 75.73 |
| SVHN | 0.07 | 0.03 | 5 | 96.87 |

Table 7: Additional training details for the experimental results in Table 4.

