# OpenReview forum: "Simultaneous Classification and Out-of-Distribution Detection Using Deep Neural Networks"
_ICLR.cc/2020/Conference — Reject_

### Official Review · AnonReviewer1 · 2019-10-20
**Official Blind Review #1**

**Rating:** 6

**Review:**

This paper proposes to tackle the problems of out-of-distribution (OOD) detection and model calibration by adapting the loss function of the Outlier Exposure (OE) technique [1]. In OE, model softmax outputs are encouraged to be uniform for OOD samples, which is enforced through the use of a KL divergence loss function. The first proposed modification in this paper is to replace KL divergence term with an L1 penalty. The second change is the addition of an L2 penalty between the maximum softmax probability and the model accuracy. Experimental results demonstrate that adding these two components increases performance over OE on standard OOD benchmarks for both vision and text domains, and also improves model calibration.

Although this paper presents some good quantitative results, I tend towards rejection in its current state. This is mainly due to the limited comparison to alternative methods, and the lack of ablation study. If these were addressed I would consider increasing my score.

Things to improve the paper:
1) Currently, one of the most commonly used benchmark methods for OOD detection is the Mahalanobis distance based confidence score (MD) [2], which, as far as I am aware, is state-of-the-art among published works. The authors claim that they do not compare to this work because it is a post-training method, and, presumably, the techniques should be doubly effective when combined. However, we do not have any proof that this is actually the case. Therefore, I think it is important to verify that the two techniques are indeed compatible, and if not, then direct comparison with MD would still be necessary.

2) In the case of confidence calibration, there is no comparison made with other calibration techniques, such as temperature scaling [3]. I think it would be good to included these for reference.

3) Since two distinct components are being added to the loss function, I think it is important to include an ablation study to identify how much each component contributes to improvements in OOD detection and confidence calibration.


Minor things to improve the paper that did not impact the score:
4) With regards to the confidence calibration loss, there is similar work by [4] which also optimizes the output of the model to make sure confidence predictions are close to the true accuracy. It may be worth citing if you think it is relevant.


Additional questions:
5) How are lambda1 and lambda2 tuned? I could not find this information in the paper.

6) How sensitive is model performance to the setting of the lambda hyperparameters? It would be nice to see a plot of lambda versus the OOD detection metrics.

7) Have you evaluated how this method performs at detecting adversarial attacks? I do not think the paper will suffer without these results, but they are certainly relevant and of interest to practitioners in this area.


References:
[1] Hendrycks, Dan, Mantas Mazeika, and Thomas G. Dietterich. "Deep anomaly detection with outlier exposure." ICLR (2018).

[2] Lee, Kimin, Kibok Lee, Honglak Lee, and Jinwoo Shin. "A simple unified framework for detecting out-of-distribution samples and adversarial attacks." NeurIPS, (2018).

[3] Guo, Chuan, Geoff Pleiss, Yu Sun, and Kilian Q. Weinberger. "On calibration of modern neural networks." In Proceedings of the 34th International Conference on Machine Learning-Volume 70, pp. 1321-1330. JMLR. org, 2017.

[4] Corbière, Charles, Nicolas Thome, Avner Bar-Hen, Matthieu Cord, and Patrick Pérez. "Addressing Failure Prediction by Learning Model Confidence." NeurIPS (2019).


### Post-Rebuttal Comments ###
I would like to thank the authors for their hard work during the rebuttal period. I think the current version of the paper is much improved over the previous version. The choice to remove the claims about calibration definitely improves the focus of the paper. The addition of the  Mahalanobis distance experiments and the ablation study also significantly strengthen the paper. However, as the other reviewers have pointed out, the novelty of the paper is quite limited since the majority of the gains come from simply swapping the KL-divergence penalty with an L1 penalty. Despite simplicity, this single change yields a significant improvement in performance for the OE algorithm, which is noteworthy. As a result, I will increase my score from a weak reject (3) to a very, very weak accept (more like a 5 than a 6).

**Experience Assessment:**

I have published one or two papers in this area.

**Review Assessment: Checking Correctness Of Derivations And Theory:**

N/A

**Review Assessment: Checking Correctness Of Experiments:**

I carefully checked the experiments.

**Review Assessment: Thoroughness In Paper Reading:**

I read the paper at least twice and used my best judgement in assessing the paper.

---

> ### Author Response · Authors · 2019-11-11
> **Answer to Reviewer #1**
>
> Thank you for your valuable comments and for taking the time and effort to review our paper. Your comments helped us to significantly improve the quality of our paper. We addressed all of them! Please check our answers below:
>
> 1)	Thank you for your valuable comment. In the initial version of our paper, we had mentioned that Mahalanobis distance-based confidence score [1] is a post-training method. At this point, we would also like to mention that that in the OOD detection task, we could classify the methods into three categories: 1) If someone assumes access to knowledge of the test distribution, then Mahalanobis distance-based classifier proposed by [1] achieves state-of-the-art results in the OOD detection task, 2) If someone does not make this assumption then the results of [2] are considered state-of-the-art, 3) if someone assumes access to no extra data during training, then the maximum softmax probability+rotation prediction is the best (please check the same argument that was made by the Anonymous Reviewer #3 of the ICLR 2020 paper in the following link: https://openreview.net/forum?id=r1g6MCEtwr).
>
> In our initially submitted version, we had proposed a loss function that beat the results of [2] in both image and text classification tasks achieving state-of-the-art results in the second category of methods, which together with the novelty of our loss function showed the contribution of our technique. Understandably, since all of the esteemed reviewers asked us to run more simulation experiments, we followed your suggestion and we tried to verify the compatibility of our method and Mahalanobis method. Therefore, in the revised version of the paper, we carried out additional image classification experiments where we trained the DNN with our proposed method and then we used the Mahalanobis method. We compared the results we obtained with those obtained by the original Mahalanobis distance-based classifier in [1]. We have added a whole section in our paper (please see Section 4.2) for this. The experimental results are presented in Table 4 of the paper where it is shown that our method combined with the Mahalanobis method outperforms the original Mahalanobis method in most of the experiments achieving state-of-the-art results even in the first category of methods for OOD detection. Furthermore, we added a discussion section (just before Section 5) where we explain these experimental results. In our opinion, these results are significant because they present a very effective and adaptable OOD detection method.
>
> 2)	Thank you for your valuable comment. Regarding the calibration term, we firmly believe that the contribution of our paper is in the field of OOD detection. The idea of calibration was one of our inspirations (but not the main one!) when we added the corresponding regularization term in our loss function. Our intent was to show that our proposed loss function can also achieve this. However, taking into account your comment and the fact that the contribution of our work is in the OOD detection field, we decided to remove both the calibration experiments and the statement of calibration improvement from the contribution of our work in the revised version of the paper. In its current state, the contribution of our paper is:
>
> a)	We propose a novel loss function for the OOD detection task consisting of two regularization terms, we try to go through all the theoretical derivation of those and we show experimentally what is the contribution of each term in both the OOD evaluation metrics as well as in the test accuracy on examples generated by $D_{in}$ as can be shown in the Table 2 in the revised version of our paper.
> b)	We achieve state-of-the-art results in the OOD detection task for the class of methods that do not make any assumption about access to the test distribution by consistently outperforming the results of [2] in both image and text classification tasks (see Tables 1,3,5 and 6 in the revised version of our paper).
> c)	We show the adaptability and the effectiveness of our method by combining it with the Mahalanobis method and by outperforming the original Mahalanobis method proposed in [1], achieving state-of-the-art results in the OOD detection task even for the first category of methods for OOD (see Section 4.2 and Table 4 in the revised version of our paper).
>
>
> 3)	Thank you for your valuable comment. We added a section (at the end of Section 4.1.2 in the revised version of our paper) and carried out additional experiments to demonstrate the effect of each regularization term both in the OOD evaluation metrics as well as in the test accuracy on examples generated by $D_{in}$. The experimental results can be found in Table 2 in the revised version of our paper.
>
> 4)	The initial arXiv version of [3] was uploaded on October 1st which is later than the submission deadline of September 25th for the ICLR paper. However, we found this work relevant and we cited it accordingly.

---

> > ### Author Response · Authors · 2019-11-11
> > **Answer to Reviewer #1 (Continued)**
> >
> > 5)	Thank you for raising this. At the end of page 5 and at the end of section 4.1.3 (for image and text classification experiments, respectively) in the revised version of our paper, we mention that we are using a separate validation dataset similar to [2]. Note that $D_{out}^{val}$ and $D_{out}^{test}$ are disjoint. The detailed description of the validation data has also been added in Appendices A.3 and B.2 for image and text classification experiments, respectively.
> >
> > 6)	Thank you for this comment. In all of our experiments (image classification, text classification, combination of our method with the Mahalanobis distance-based confidence score etc.), the values of the lambda hyperparameters were in the range [0.03, 0.1] (as also mentioned in the Experiments section of our paper) which shows that those values are general enough to achieve state-of-the-art results in all the experiments. Therefore, the answer to your question is that the model performance is not sensitive to the setting of the lambda hyperparameters. At this point, we do not have a plot demonstrating this but we have a 3D plot demonstrating the test accuracy of the model on in-distribution data for different values of lambda_1 and lambda_2. Note that since plot comes from an early version of our work, it has been produced by training a 6-layer NN in the MiniImagenet dataset and both regularization terms use $l_2$ norms instead of $l_2$ and $l_1$ that we propose in our paper. Please let us know if you think that this plot should be included in our paper.
> >
> > 7)	This is a very interesting direction that calls for another paper and we will definitely evaluate our method against adversarial attacks in a future work.
> >
> >
> > [1] Lee, Kimin, Kibok Lee, Honglak Lee, and Jinwoo Shin. "A simple unified framework for detecting out-of-distribution samples and adversarial attacks." NeurIPS, (2018).
> > [2] Hendrycks, Dan, Mantas Mazeika, and Thomas G. Dietterich. "Deep anomaly detection with outlier exposure." ICLR (2019).
> > [3] Corbière, Charles, Nicolas Thome, Avner Bar-Hen, Matthieu Cord, and Patrick Pérez. "Addressing Failure Prediction by Learning Model Confidence." NeurIPS (2019)

---

> > > ### Author Response · Authors · 2019-11-12
> > > **UPDATE**
> > >
> > > Dear Reviewers,
> > >
> > > We updated the experimental results in Table 4 (related to the SVHN experiment) of our paper where we compare the Mahalanobis distance-based classifier [1] versus the combination of our method with the Mahalanobis method. It is obvious that the combination of our method with the Mahalanobis method consistently outperforms the original Mahalanobis method proposed in [1] and achieves state-of-the-art results in the OOD detection task (even for the first category of methods for OOD detection). This is not surprising since even our method alone (which makes NO assumption about access to the test distribution data as the Mahalanobis method does) achieves almost 100% performance when SVHN is assumed to be generated by $D_{in}$ as can be seen by the experimental results presented in Table 1 in the revised version of our paper. We believe that these experiments show both the adaptability and the effectiveness of our method for OOD detection.
> > > We are currently cleaning the code. The full code will be updated by tomorrow.
> > >
> > >
> > > [1] Lee, Kimin, Kibok Lee, Honglak Lee, and Jinwoo Shin. "A simple unified framework for detecting out-of-distribution samples and adversarial attacks." NeurIPS, (2018).

---

### Official Review · AnonReviewer3 · 2019-10-21
**Official Blind Review #3**

**Rating:** 1

**Review:**

This work proposes a new loss function to train the network with Outlier Exposure(OE) [1] which leads to better OOD detection compared to simple loss function that uses KL divergence as the regularizer for OOD detection. The new loss function is the cross entropy plus two more regularizers which are : 1) Average ECE (Expected Calibration Error) function to calibrate the model and 2) absolute difference of the network output to $1/K$ where $K$ is the number of tasks. The second regularizer keeps the softmax output of the network uniform for the OE samples. They show adding these new regularizers to the cross-entropy loss function will improve the Out-distribution detection capability of networks more than OE method proposed in [1] and the baseline proposed in [2].


Pros:
The paper is written clearly and the motivation of designed loss functions are explained well.

Cons:
1- The level of contributions is limited.

2- The variety of comparison is not enough. The authors did not show how the approach is working in compared to the other OOD methods like ODIN[3] and the proposed method in [4].

3- The experiments are not supporting the idea. First, the paper claims that the KL is not a good regularizer for OOD detection as it is not a distance metric. But there is no experiment or justification in the paper that supports why this claim is true. Then the second contribution claims that the calibration term that is added to the loss function improves the OOD detection as well as calibration in the network, but the experiments are not designed to show the impact of  each regularizer term separately in improving the OOD detection rate.  Figure 2 also does not depict any significant conclusion. It only shows that the new loss function makes the network more calibrated than the naive network. This phenomenon was reported before in [1]. It would be better if the paper investigated the relation between the calibration and OOD detection by designing more specific experiments for calibration section.

Overall, I think the paper should be rejected as the contributions are limited and are not aligned with the experiments.

References
[1]Hendrycks, Dan, Mantas Mazeika, and Thomas G. Dietterich. "Deep Anomaly Detection with Outlier Exposure." arXiv preprint arXiv:1812.04606 (2018).

[2] A Baseline for Detecting Misclassified and Out-of-Distribution Examples in Neural Networks, ICLR2016.

[3] Liang, Shiyu, Yixuan Li, and Rayadurgam Srikant. "Enhancing the reliability of out-of-distribution image detection in neural networks." arXiv preprint arXiv:1706.02690 (2017).

[4] Lee, Kimin, et al. "A simple unified framework for detecting out-of-distribution samples and adversarial attacks." Advances in Neural Information Processing Systems. 2018.

**Experience Assessment:**

I have read many papers in this area.

**Review Assessment: Checking Correctness Of Derivations And Theory:**

I did not assess the derivations or theory.

**Review Assessment: Checking Correctness Of Experiments:**

I carefully checked the experiments.

**Review Assessment: Thoroughness In Paper Reading:**

I read the paper at least twice and used my best judgement in assessing the paper.

---

> ### Author Response · Authors · 2019-11-11
> **Answer to Reviewer #3**
>
> Thank you for your valuable comments since they helped us significantly improve the quality of our paper. We addressed all of your concerns and comments. Please see our answer below:
>
> 1)	At first, we would like to mention that in the OOD detection task, we could classify the methods into three categories: 1) If someone assumes access to knowledge of the test distribution, then Mahalanobis distance-based classifier proposed by [1] achieves state-of-the-art results in the OOD detection task, 2) If someone does not make this assumption then the results of [2] are considered state-of-the-art, 3) if someone assumes access to no extra data during training, then the maximum softmax probability+rotation prediction is the best (please check the same argument that was made by the Anonymous Reviewer #3 of the ICLR 2020 paper in the following link: https://openreview.net/forum?id=r1g6MCEtwr).
>
> In our initially submitted version, we had proposed a loss function that beat the results of [2] in both image and text classification tasks achieving state-of-the-results in the second category of methods which together with the novelty of our loss function showed the contribution of our technique. Understandably, all of the esteemed reviewers asked us to run more simulation experiments and compare our results with more methods, which resulted in significant improvements in the paper and proved the merits of our method. In the initial version of our paper, we had cited the seminal work of [1] stating that Mahalanobis distance-based classifier is a post-training method and it is not directly comparable to ours. Fortunately, the Anonymous Reviewer #1 of our paper brought to our attention that we should verify whether the two methods are indeed compatible. Therefore, in the revised version of the paper, we ran additional image classification experiments where we trained the DNN with our proposed method and then we used the Mahalanobis method. We compared the results we obtained  with those obtained by the original Mahalanobis distance-based classifier in [1]. We have dedicated a whole section in our paper (please see Section 4.2) to this combination. The experimental results are presented in Table 4 of the paper where it is shown that our method combined with the Mahalanobis method outperforms the original Mahalanobis method in most of the experiments achieving state-of-the-art results even in the first category of methods for OOD detection. Furthermore, we added a discussion section (just before Section 5) where we explain these experimental results. In our opinion, these results are significant because they present a very effective and flexible OOD detection method. Additionally, in order to further demonstrate the insights of our method we ran some additional experiments for CIFAR-10 and CIFAR-100 datasets where we show what the contribution of each regularization term in the OOD evaluation metrics is, as well as in the test accuracy of the DNN in examples generated by $D_{in}$ (Please check the results of these experiments in Table 2.). In its current state, the contributions of our paper are:
>
> 1)	We propose a novel loss function for the OOD detection task consisting of two regularization terms, we thoroughly explain the theoretical and mathematical foundations of those regularization terms and we show experimentally the contribution of each term in both the OOD evaluation metrics as well as in the test accuracy on examples generated by $D_{in}$.
> 2)	We achieve state-of-the-art results in the OOD detection task for the class of methods that do not make any assumption about access to the test distribution by consistently outperforming the results of [2] in both image and text classification tasks (see Tables 1,3,5 and 6 in the revised version of our paper).
> 3)	We show the adaptability and the effectiveness of our method by combining it with the Mahalanobis method and by outperforming the original Mahalanobis method proposed in [1], achieving state-of-the-art results in the OOD detection task even for the first category of methods for OOD (see Section 4.2 and Table 4 in the revised version of our paper).
>
> We believe that the aforementioned arguments explain the novelty and the contribution of our paper in the field of OOD detection.

---

> > ### Author Response · Authors · 2019-11-11
> > **Answer to Reviewer #3 (Continued)**
> >
> > 2)	As mentioned in our previous answer, there are 3 categories of methods in the field of OOD detection (please check the same argument that was made by the Anonymous Reviewer #3 of the ICLR 2020 paper in the following link: https://openreview.net/forum?id=r1g6MCEtwr). Since our method assumes NO access to the test distribution and therefore belongs to the second category of methods, we compared it with the previous state-of-the-art method [2] and we showed  that our method consistently outperforms the results in [2] for both image and text classification tasks and achieved state-of-the-art results in the second category of methods for OOD detection.
> >
> > Regarding the ODIN method [3] which belongs to the first category of methods for OOD detection, it has been shown in [1], that the Mahalanobis method (which also belongs to the first category of methods for OOD detection) consistently outperforms the results of [3]. On the other hand, in the initial version of our paper, we had cited the seminal work of [1] stating that Mahalanobis distance-based classifier is a post-training method and it is not directly comparable to ours. Fortunately, the Anonymous Reviewer #1 of our paper brought to our attention that we should verify whether the two methods are indeed compatible. Therefore, in the revised version of the paper, we carried out additional image classification experiments where we trained the DNN with our proposed method and then we used the Mahalanobis method. We compared the results we obtained with those obtained by the original Mahalanobis distance-based classifier in [1]. We have added a whole section in our paper (please see Section 4.2) for this. The experimental results are presented in Table 4 of the paper where it is shown that our method combined with the Mahalanobis method outperforms the original Mahalanobis method in most of the experiments achieving state-of-the-art results even in the first category of methods for OOD detection. Furthermore, we added a discussion section (just before Section 5) where we explain these experimental results. In our opinion, these results are significant because they present a very effective and flexible OOD detection method.

---

> > > ### Author Response · Authors · 2019-11-11
> > > **Answer to Reviewer #3 (Continued)**
> > >
> > > 3)	Thank you for your valuable comment. In the initial version of our paper, we did not mention that KL is not a good regularizer for OOD detection. KL has achieved and will continue achieving great success in the machine learning field. We had mentioned that it does not satisfy the symmetry and the triangular inequality properties as required by a metric measure which is a known result. Our initial intent was to present that someone can use the $l_1$ norm to measure the distance between the output distribution produced by softmax and the uniform distribution which satisfies all mathematical properties of a distance metric (it is positive definite, symmetric and satisfies the triangular inequality property) and can also achieve great results (as KL had already done in [1], [2]) in the OOD detection task as can be shown in Tables 1,3,5 and 6 in the revised version of our paper.
> > >
> > > Regarding the calibration term, we firmly believe that the contribution of our paper is in the field of OOD detection. The idea of calibration was one of our inspirations (but not the main one!) when we added the corresponding regularization term in our loss function. As you correctly mentioned in your comment, the phenomenon that OE can improve the calibration of a naïve network has already been shown in [2]. Our intent was to show that our proposed loss function can also achieve this. However, taking into account your comment and the fact that the contribution of our work is in the OOD detection field, we decided to remove both the calibration experiments and the statement of calibration improvement from the contribution of our work in the revised version of the paper.
> > > To demonstrate the impact of each regularizer in the OOD evaluation metrics as well as in the accuracy of the DNN on the test examples of the in-distribution data, we ran some additional experiments for both CIFAR-10 and CIFAR-100 and by using 8 out-of-distribution datasets to validate our experiments. The results can be found in Table 2 of our paper. As we also mention in the corresponding section in the revised version of our paper, the experimental results of Table 2 validate that the combination of the two regularization terms of our proposed loss function not only improves the OOD detection performance of the DNN but also improves its accuracy on the test examples of D_{in} compared to the case where $\lambda_1$ = 0. This Table also demonstrates that our method can significantly improve the OOD detection performance of the DNN compared to the case where only the cross-entropy loss is minimized at the expense of only an insignificant degradation in the test accuracy of the DNN on examples generated by $D_{in}$.
> > >
> > > [1] Lee, Kimin, Kibok Lee, Honglak Lee, and Jinwoo Shin. "A simple unified framework for detecting out-of-distribution samples and adversarial attacks." NeurIPS, (2018).
> > > [2] Hendrycks, Dan, Mantas Mazeika, and Thomas G. Dietterich. "Deep anomaly detection with outlier exposure." ICLR (2019).
> > > [3] Liang, Shiyu, Yixuan Li, and Rayadurgam Srikant. "Enhancing the reliability of out-of-distribution image detection in neural networks." arXiv preprint arXiv:1706.02690 (2017).

---

> > > > ### Author Response · Authors · 2019-11-12
> > > > **UPDATE**
> > > >
> > > > Dear Reviewers,
> > > >
> > > > We updated the experimental results in Table 4 (related to the SVHN experiment) of our paper where we compare the Mahalanobis distance-based classifier [1] versus the combination of our method with the Mahalanobis method. It is obvious that the combination of our method with the Mahalanobis method consistently outperforms the original Mahalanobis method proposed in [1] and achieves state-of-the-art results in the OOD detection task (even for the first category of methods for OOD detection). This is not surprising since even our method alone (which makes NO assumption about access to the test distribution data as the Mahalanobis method does) achieves almost 100% performance when SVHN is assumed to be generated by $D_{in}$ as can be seen by the experimental results presented in Table 1 in the revised version of our paper. We believe that these experiments show both the adaptability and the effectiveness of our method for OOD detection.
> > > > We are currently cleaning the code. The full code will be updated by tomorrow.
> > > >
> > > >
> > > > [1] Lee, Kimin, Kibok Lee, Honglak Lee, and Jinwoo Shin. "A simple unified framework for detecting out-of-distribution samples and adversarial attacks." NeurIPS, (2018).

---

### Official Review · AnonReviewer2 · 2019-10-23
**Official Blind Review #2**

**Rating:** 3

**Review:**

The paper considers the problem of out-of-distribution detection in the context of image and text classification with deep neural networks. The proposed method is based on [1], where cross-entropy between a uniform distribution and the predictive distribution is maximized on the out-of-distribution data during training. The authors propose two simple modifications to the objective. The first modification is to replace cross-entropy between predictive and uniform distributions with an l1-norm. The second modification is to add a separate loss term that encourages the average confidence on training data to be close to the training accuracy. The authors show that these modifications improve results compared to [1] on image and text classification.

There are a few concerns I have for this paper. The main issue is that I am not sure if the level of novelty is sufficient for ICLR. The contributions of the paper consist of a new loss term and a modification of the other loss term in OE [1]. At the same time, the paper achieves an improvement over OE consistently on all the considered problems. Given the limited novelty, experiments aimed at understanding the proposed modification would strengthen the paper. Right now I am leaning towards rejecting the paper, but if authors add more insight into why the proposed modifications help, or provide a strong rebuttal, I may update my score.

I discuss the other less general concerns below.
1. I believe the presentation of the method in Section 3 is suboptimal. The authors start with presenting a constrained minimization problem, then convert it to a problem with Lagrange multipliers, then modify the problem in an ad hoc way (adding a square and a norm without any mathematical reason), to get the standard form of loss with regularizers. The presentation would be much cleaner, and wouldn’t lose anything if the authors directly presented the loss with regularizers. Furthermore, in the Lagrange multiplier view the Lagrange multipliers are not hyper-parameters, they are dual variables, and the optimization problem shouldn’t be just with respect to theta, we need to find a stationary point with respect to both lambda and theta.

2. The motivation for changing the distance measure between the predictive distribution on outlier data and the uniform distribution is unclear. The authors state multiple times that KL is not a distance metric, but it isn’t clear why this is important. KL is commonly used as a measure of distance between distributions. One could also use symmetrized KL in order to get a distance metric that is similar to KL. I am not opposed to just using l1-norm because it performs better, but if the switch of distance measures is listed as one of the two main methodological contributions, I believe more insight needs to be provided for why it helps.

3. The motivation for the other loss term which is enforcing calibration of uncertainty on train data is also not very clear. At least in image classification, strong networks typically achieve perfect accuracy (or close to that) on the train set, and then the proposed loss term would basically push the predictive confidence on all training data to 1, which is already enforced by the standard cross-entropy loss. Does outlier exposure prevent the classifier from getting close to 100% accuracy on train? What is the train accuracy for the experiments on CIFAR-10, CIFAR-100 and SVHN?

4.  While the authors report accuracy of out-of-distribution detection in the experiments,  they don’t report the accuracy of the actual classifier on in-distribution data. Is this accuracy similar for the proposed method and OE? Is the accuracy also similar for the proposed method and baseline for the experiment in section 4.4?

5. The method is only being compared to the OE method of [1]. Why is this comparison important, and are there other methods that the authors could compare against?

[1] Deep Anomaly Detection with Outlier Exposure. Dan Hendrycks, Mantas Mazeika, Thomas Dietterich

**Experience Assessment:**

I have read many papers in this area.

**Review Assessment: Checking Correctness Of Derivations And Theory:**

N/A

**Review Assessment: Checking Correctness Of Experiments:**

I assessed the sensibility of the experiments.

**Review Assessment: Thoroughness In Paper Reading:**

I read the paper at least twice and used my best judgement in assessing the paper.

---

> ### Author Response · Authors · 2019-11-11
> **Answer to Reviewer #2**
>
> Thank you for your response and your valuable feedback since we firmly believe that it helped us to improve the quality of our paper.
>
> We took into consideration all of your comments. Since you initially wrote a general comment, let us first answer to this one and subsequently, we are going to answer all the other less general concerns. Please see our answer below:
>
> In the revised version of the paper, we tried to better explain the insights of our method both theoretically and experimentally. At first, we would like to mention that in the OOD detection task, we could classify the methods into three categories: 1) If someone assumes access to knowledge of the test distribution, then Mahalanobis distance-based classifier proposed by [1] achieves state-of-the-art results in the OOD detection task, 2) If someone does not make this assumption then the results of [2] are considered state-of-the-art, 3) if someone assumes access to no extra data during training, then the maximum softmax probability+rotation prediction is the best (please check the same argument that was made by the Anonymous Reviewer #3 of the ICLR 2020 paper in the following link: https://openreview.net/forum?id=r1g6MCEtwr).
>
> In our initially submitted version, we had proposed a loss function that beat the results of [2] in both image and text classification tasks achieving state-of-the-results in the second category of methods which together with the novelty of our loss function showed the contribution of our technique. Understandably, all of the esteemed reviewers asked us to run more simulation experiments and compare our results with more methods, which resulted in significant improvements in the paper and proved the merits of our method. In the initial version of our paper, we had cited the seminal work of [1] stating that Mahalanobis distance-based classifier is a post-training method and it is not directly comparable to ours. Fortunately, the Anonymous Reviewer #1 of our paper brought to our attention that we should verify whether the two methods are indeed compatible. Therefore, in the revised version of the paper, we ran additional image classification experiments where we trained the DNN with our proposed method and then we used the Mahalanobis method. We compared the results we obtained  with those obtained by the original Mahalanobis distance-based classifier in [1]. We have dedicated a whole section in our paper (please see Section 4.2) to this combination. The experimental results are presented in Table 4 of the paper where it is shown that our method combined with the Mahalanobis method outperforms the original Mahalanobis method in most of the experiments achieving state-of-the-art results even in the first category of methods for OOD detection. Furthermore, we added a discussion section (just before Section 5) where we explain these experimental results. In our opinion, these results are significant because they present a very effective and flexible OOD detection method. Additionally, in order to further demonstrate the insights of our method we ran some additional experiments for CIFAR-10 and CIFAR-100 datasets where we show what the contribution of each regularization term in the OOD evaluation metrics is, as well as in the test accuracy of the DNN in examples generated by $D_{in}$ (Please check the results of these experiments in Table 2.). In its current state, the contributions of our paper are:
>
> 1)	We propose a novel loss function for the OOD detection task consisting of two regularization terms, we thoroughly explain the theoretical and mathematical foundations of those regularization terms and we show experimentally the contribution of each term in both the OOD evaluation metrics as well as in the test accuracy on examples generated by $D_{in}$.
> 2)	We achieve state-of-the-art results in the OOD detection task for the class of methods that do not make any assumption about access to the test distribution by consistently outperforming the results of [2] in both image and text classification tasks (see Tables 1,3,5 and 6 in the revised version of our paper).
> 3)	We show the adaptability and the effectiveness of our method by combining it with the Mahalanobis method and by outperforming the original Mahalanobis method proposed in [1], achieving state-of-the-art results in the OOD detection task even for the first category of methods for OOD (see Section 4.2 and Table 4 in the revised version of our paper).
>
> We believe that the aforementioned arguments explain the novelty and the contribution of our paper in the field of OOD detection.

---

> > ### Author Response · Authors · 2019-11-11
> > **Answer to Reviewer #2 (Continued)**
> >
> > Less general concerns:
> >
> > 1)	Thank you for your valuable comment. We would like to keep the presentation of our method as it is because we think that presenting the theoretical/mathematical derivation of our method helps the reader to better understand its insights. Regarding the Lagrange multipliers, we totally agree with your comment that the optimization in the Lagrangian is with respect to both $/theta$ and $/lambda$. But in machine learning applications, usually we do not optimize over lambdas and we decide their values using a validation technique. Still, in many Machine Learning references (e.g. [8]), those hyperparameters are considered to have come from a Lagrange optimization setting. Therefore, we addressed your comment by explaining this after equation (2). As far as it concerns using the $l_2$ norm in the first regularizer and the $l_1$ norm in the second regularizer (i.e when we transition from equation (2) to equation (3)), there exists a mathematical reason behind it and we try to explain this in our paper. At first, as also mentioned before presenting equation (3) in our paper, we need to use some norms in order to prevent our loss function from taking negative values. The reason for which we specifically picked $l_2$ and $l_1$ is explained in the last paragraph of Section 3 in the revised version of our paper. Let us explain it further here. The purpose of the second regularizer is to create a uniform distribution in the output of the DNN for the OOD data. For an image classification task where the initial number of classes for in-distribution is K=10, the uniform distribution for the OOD data can be achieved if ALL the prediction probabilities of the softmax layer for the OOD data is 0.1. It is generally known (see for example the seminal work of [3]) that deep neural networks may make predictions for out-of-distribution (OOD) examples with high confidence. That means that initially, in a simple example with only 3 OOD data points, the DNN might have confidence for OOD data 0.4, 0.7 and 0.9, respectively. As we mentioned before, to achieve uniform distribution at the output of the softmax layer of the DNN for the OOD data, we need to bring those values to 0.1 (for K=10). The $l_1$ is the only norm that will uniformly attract those values (0.4, 0.7 and 0.9) towards 0.1. This explains the use of the $l_1$ norm as the second regularizer of our method. Let us now explain the use of the $l_2$ norm as the first regularizer in our method. Methods like [1] or [2] which try to create a uniform distribution at the output of the softmax layer of a DNN for the OOD data, will inevitably make the DNN have low confidence for some predictions which are related to in-distribution examples creating a small overlap between in- and out-of-distribution examples at the low softmax probability levels, let’s say around 0.13 according to the previous example. If the DNN has a test accuracy on in-distribution data around 85%, the use of l_2 norm will put more effort to attract the in-distribution examples with a prediction probability far away from 0.85. In our case, those examples will be the ones with maximum prediction probabilities around 0.13. This helps us to better distinguish in- and out-of-distribution data at the low softmax probability levels as it is also explained in the last paragraph of Section 3 in the revised version of our paper. We think that the above explanation provides the theoretical/mathematical arguments needed to support our method and the use of specific norms.
> >
> > 2)	First, we would like to say that KL has achieved great success in the machine learning field. Our initial intent was to present that someone can use the $l_1$ norm to measure the distance between the output distribution produced by softmax and the uniform distribution, which satisfies mathematical properties of a distance metric (is positive definite and symmetric and satisfies triangular inequality) and can also achieve great results in the OOD detection task as can be shown in Tables 1,3,5 and 6. However, since the main contribution of our paper is the OOD detection task, we decided to remove this statement from our main contributions. We totally agree with the Reviewers that it is hard to attribute the success of $l_1$ norm in our paper to this particular argument.

---

> > > ### Author Response · Authors · 2019-11-11
> > > **Answer to Reviewer #2 (Continued)**
> > >
> > > 3)	Thank you for raising this with your comment. The theoretical/mathematical arguments that support the existence of this regularization term in our loss function have been addressed in our answer to your 1st less general concern. So, let us explain why this term should push the maximum probabilities for in-distribution data around $A_{tr}$. The motivation for pushing the maximum probabilities for in-distribution data around $A_{tr}$ is better explained both in the Introduction Section of our paper as well as in the Section 3. It is known from the seminal work of [3] that deep neural networks can make predictions for out-of-distribution (OOD) examples with high confidence. In [4], the authors found that high confidence predictions consist a symptom of overfitting. They also make the calibration of neural networks difficult as in [5], the authors observed that modern neural networks are miscalibrated by experimentally showing that the average confidence of deep neural networks is usually much higher than their accuracy. Additionally, viewing the knowledge of a model as the class conditional distribution it produces over outputs given an input as proposed by [6], the authors of [7] used the entropy of this conditional distribution as a regularization term in order to penalize the confident predictions of a neural network. These arguments that are incorporated in our paper, motivated us to define the regularization term which minimizes the Euclidean distance between the training accuracy of a DNN and its average confidence in its predictions on the training set.
> > >
> > > Regarding the accuracy of the DNN on the test examples generated by $D_{in}$, we report these results in Table 2 of our paper, where we make a comparison with the case where only the cross-entropy loss is used during training. It had been observed in [2] that OE can have a small degradation in the test accuracy on examples generated by $D_{in}$ but we experimentally showed that this degradation is insignificant especially with the additional use of the first regularization term as shown in Table 2 of our paper.
> > >
> > > 4)	We addressed this comment by adding Table 2 in our paper and by our previous answer.
> > >
> > > 5)	As also explained in our answer to your general concern, if someone makes NO assumption about access to the test distribution (which is the case for our method), then OE method of [2] is state-of-the-art. We showed that our method consistently outperforms [2] in both image and text classification tasks by using a huge variety of datasets. In the revised version of our paper, we made the comparison with the Mahalanobis distance-based classifier [1] which is the state-of-the-art method if someone makes an assumption about access to the test distribution. As we also explained in our answer in your general concern above, the newly added Section 4.2 in our paper shows how our method combined with the Mahalanobis method outperforms the original Mahalanobis method proposed by [1] and achieves state-of-the-art results even for this category of methods for OOD detection.
> > >
> > >
> > > [1] Lee, Kimin, Kibok Lee, Honglak Lee, and Jinwoo Shin. "A simple unified framework for detecting out-of-distribution samples and adversarial attacks." NeurIPS, (2018).
> > > [2] Hendrycks, Dan, Mantas Mazeika, and Thomas G. Dietterich. "Deep anomaly detection with outlier exposure." ICLR (2019).
> > > [3] Anh Mai Nguyen, Jason Yosinski, and Jeff Clune. “Deep neural networks are easily fooled: High confidence predictions for unrecognizable images.” CVPR, 2015.
> > > [4] Christian Szegedy, Vincent Vanhoucke, Sergey Ioffe, Jonathon Shlens, and Zbigniew Wojna. Rethinking the inception architecture for computer vision. arXiv preprint arXiv: 1512.00567, 2015.
> > > [5] Chuan Guo, Geoff Pleiss, Yu Sun, and Kilian Q. Weinberger. On calibration of modern neural networks. In International Conference on Machine Learning (ICML), 2017.
> > > [6] Geoffrey Hinton, Oriol Vinyals, and Jeff Dean. Distilling the knowledge in a neural network. arXiv preprint arXiv:1503.02531, 2015.
> > > [7] Gabriel Pereyra, George Tucker, Jan Chorowski, Lukasz Kaiser, and Geoffrey E. Hinton. Regularizing neural networks by penalizing confident output distributions. In International Conferece on Learning Representations (ICLR), 2017.
> > > [8] Trevor Hastie, Robert Tibshirani, and Jerome Friedman. The Elements of Statistical Learning, Springer, NY, USA, 2001.

---

> > > > ### Author Response · Authors · 2019-11-12
> > > > **UPDATE**
> > > >
> > > > Dear Reviewers,
> > > >
> > > > We updated the experimental results in Table 4 (related to the SVHN experiment) of our paper where we compare the Mahalanobis distance-based classifier [1] versus the combination of our method with the Mahalanobis method. It is obvious that the combination of our method with the Mahalanobis method consistently outperforms the original Mahalanobis method proposed in [1] and achieves state-of-the-art results in the OOD detection task (even for the first category of methods for OOD detection). This is not surprising since even our method alone (which makes NO assumption about access to the test distribution data as the Mahalanobis method does) achieves almost 100% performance when SVHN is assumed to be generated by $D_{in}$ as can be seen by the experimental results presented in Table 1 in the revised version of our paper. We believe that these experiments show both the adaptability and the effectiveness of our method for OOD detection.
> > > > We are currently cleaning the code. The full code will be updated by tomorrow.
> > > >
> > > >
> > > > [1] Lee, Kimin, Kibok Lee, Honglak Lee, and Jinwoo Shin. "A simple unified framework for detecting out-of-distribution samples and adversarial attacks." NeurIPS, (2018).

---

### Decision · Program_Chairs · 2019-12-19

**Decision:**

Reject

**Comment:**

The paper proposes a method for out-of-distribution (OOD) detection for neural network classifiers.

The reviewers raised several concerns about novelty, choice of baselines and the experimental evaluation. While the author rebuttal addressed some of these concerns, I think the paper is still not ready for acceptance as is.

I encourage the authors to revise the paper and resubmit to a different venue.